# Hematologic profile of Amazon river dolphins *Inia geoffrensis* and its variation during acute capture stress

Daniela M. D. de Mello 👤*, Vera M. F. da Silva

Aquatic Mammals Laboratory, National Institute of Amazonian Research, Manaus, Amazonas, Brazil

\* danielamello@hotmail.com

## Abstract

Hematological values are of primary importance when investigating the health and physiological status of populations as they reflect the biological equilibrium of aquatic ecosystems. The objectives of this study are to produce baseline values for hematological parameters of the Amazon River dolphin (*Inia geoffrensis*), as well as to investigate significant variations according to sex, age, reproductive status and stress level. One-hundred-and-ten dolphins from Mamirauá Sustainable Development Reserve (3˚3'S, 64˚51'W), Central Amazon, Brazil, were live captured and sampled in November 2004 and 2005. Further, the means, standard deviations, minimum and maximum values and reference values (90% CI) were calculated. Correlations were performed to assess the relationships among blood values and cardiac rate (CR), respiratory frequency (RF), handling time and level of stress. No significant differences were found between sexes. Also, no differences occurred among pregnant and non-pregnant females, pregnant females and adult males or non-pregnant females and adult males. Calves had a higher white blood cell (WBC) count, and the neutrophil and lymphocyte absolute counts were significantly higher in calves than adults. The level of stress determined by empirical observation positively correlated with the WBC, neutrophil, lymphocyte and monocyte absolute counts and CR and RF. It was found that less stressed animals tend to present lower platelet counts and lower CR. The handling time of the dolphins was positively correlated with hematocrit (Hct), red blood cells (RBC) and Hb level. The hematological and physiological parameters varied according to time of handling and proved to be a good bioindicator of acute stress in Amazon River dolphins. The data provided here can complement long-term monitoring and identify the early warning indicators of health problems at the population level.

## Introduction

Methods and techniques that advance the understanding of the physiological responses to the environment are becoming commonplace in the ecological studies of wild animals [1]. The health status of a population can be assessed by blood analysis of wild individuals [2,3]. The baseline data could then be used to evaluate the impact of environmental or anthropogenic health threats; *e.g.*, pollutants [4]; firestorms [5]; algal toxins and cold temperatures [6] and noise stress [7]. Health

---

**Data Availability Statement:** All relevant data are within the manuscript and its Supporting Information files.

**Funding:** This Project was financially supported by Laboratório de Mamíferos Aquáticos/ National

---

Institute for Amazonian Research / PPI number 1-3920, Coordenação de Aperfeiçoamento de Pessoal de Nível Superior (CAPES) which provided the DMDM master's scholarship, and the financial sponsors of the Projeto Boto. The funders had no role in study design, data collection and analysis, decision to publish, or preparation of the manuscript.

**Competing interests:** The authors have declared that no competing interests exist.

assessments have been carried out on netted open ocean wild populations of dolphins [8], as well as in capture-release programs on coastal species [3,9,10] and have successfully provided blood parameter baselines for different species. The applicability of this kind of study can be perceived in a study where blood alterations were observed in bottlenose dolphins (*Tursiops truncatus*) after an oil spill which occurred due to the explosion of an offshore platform [11]. Blood values had been previously established for the species in Sarasota Bay population, Florida, USA, and had served as a reference to evaluate the Barataria Bay, Louisiana, USA, health population [12][13].

Distinct blood values can be observed in dolphins that endure different physiological states. Younger cetaceans may present different blood values than older individuals [9,14], including higher white blood cell (WBC) counts [15] [16]. Females may present higher haematocrit values than males [9], while pregnant females may have lower hematocrit (Hct) than males and non-pregnant females [3]. Alterations in blood values may also occur due to stress caused by transportation. The transportation and handling of wild animals may involve different steps from capturing to sampling, and the influence/duration of each step may be reflected in distinct blood alterations [8]. Given the significant differences that may occur in blood values according to sex, age, occurrence of pregnancy [15] and stress level, these potential differences should be investigated before establishing the baseline blood parameters.

The Amazon River dolphin or boto (*Inia geoffrensis*) is endemic to the Amazonian biome [17] and has been recently classified as an endangered by the International Union for Conservation of Nature's Red List of Threatened Species [18]. These dolphins have been actively fished in different regions of Amazon to serve as bait for piracatinga (*Calophysus macropterus*) fishery, which led to a significant population decline [19,20]. Although some studies have analyzed captive Amazon River dolphins [21,22], no comprehensive health assessments have been conducted in the vast area of Amazonia so far. It is likely that Amazon River dolphins (*I. geoffrensis*), as a top predator, may reflect the effects of natural or anthropogenic stressors. Being long-lived and long-term residents of the Mamirauá Sustainable Development Reserve (RDSM) in the Amazon River, these dolphins can serve as important sentinels of an area that had very limited human interference in the past.

The current study was designed primarily to establish baseline values for the conventional hematological parameters of Amazon River dolphins. Further, variations in certain blood parameters according to gender, age, reproductive status and stress level are explored. The study also relates stress level and capture time to the variation in the hematological parameters. These data may be used to complement the long-term monitoring efforts of the dolphin population at Mamirauá [23,24] and to identify early warning indicators of potential long-term health problems at the population level.

## Materials and methods

### Animal ethics

The study was approved by the Committee on the Ethics of Animal Experiments of the National Institute for Amazonian Research. Every possible effort was made to ensure the safety and health of animals and personnel throughout the capture, examination and blood sampling procedure. No anesthesia, euthanasia or surgical procedure was used in this study. The dolphins were constantly provided sufficient shade and were kept wet while their behavioral and respiratory patterns were being closely monitored.

### Study site

The study was conducted at RDSM (3˚3'S, 64˚51'W), an area of 11,000 km$^2$ that is located between the Solimões and Japurá Rivers in the central Amazon region of Brazil. The specific

localities from which the animals were captured include the Boca do Mamirauá and Paraná do Apara. A laboratory was set up at a floating research station, which was equipped with a gaso-line-powered generator and was supplied with all the necessary equipment and reagents.

## Animal capture and handling

The dolphins were captured using a previously perfected seining technique and then moved to a shallow beach [25] before being transported by boat to a nearby floating laboratory. The total time of capture and handling were registered from the moment the animals were encircled and covered with the small mesh net to their release in the proximity of the floating research station after the sampling procedures. The number and behavior of the encircled animals varied according to environmental conditions. A net of 12 meters in height was used to completely block a channel (maximum 10 meters depth), and the isolated area was subdivided by the deployment of more nets. The dolphins were brought together by seining them into a gradually shelving shoreline with a separate small mesh net.

The animals that exhibited any external signs of ill health were excluded from subsequent analyses. Age classes were determined according to animal's total body length and sexual maturity [26]. Body color, scarring and skin lesions were also used to identify the age class of the 54 males and 56 females of the present study [27] (Table 1). Pregnancy was determined through ultrasound examinations (SonoSite Vet180plus, SonoSite Inc., Bothell, Washington, 98021–3904, USA) that were conducted on 33 adult females.

This work was carried out after successive permits were granted to Projeto Boto by IBAMA/SISBIO (Brazilian National Environmental Agency) number 13157–1 and after it satisfied its criteria for ethics and animal welfare.

## Sampling and blood analysis

Most of blood samples were collected from 110 relatively calm dolphins while they were manually restrained at the floating research station. Around 15 ml of blood was drawn from the superficial caudal peduncle vessel using a 19-gauge, 1.9 cm long butterfly catheter into a 20 ml syringe (Becton Dickinson Indústrias Cirúrgicas Ltda., São Paulo, São Paulo, 04717–004, Brazil) and was transferred to 5ml ethylenediaminetetracetic acid (EDTA) Vacutainer tubes (Becton Dickinson Indústrias Cirúrgicas Ltda., São Paulo, São Paulo, 04717–004, Brazil). After filling the tubes, they were gently rocked and then refrigerated within a maximum period of 20 mins following initial blood collection. The remaining blood was transferred into tubes with no anticoagulant and allowed to clot in ambient temperature to be used in other studies.

Manual hematological techniques were employed by a single trained individual for all the measurements. The red blood cell (RBC) and WBC counts were determined using Hayem and Turck (acetic acid and gentian violet) reagents [28], respectively (Newprov Produtos para

**Table 1. Amazon River dolphins captured and released at RDSM, Brazil, in November 2004 and 2005.**

| Gender and age class | N | Min length (cm) | Max length (cm) |
|---|---|---|---|
| Adult male | 42 | 190 | 248 |
| Adult female | 43 | 171 | 212 |
| Juvenile male | 6 | 169 | 189 |
| Juvenile female | 7 | 151 | 166 |
| Male calf | 6 | 119 | 152 |
| Female calf | 6 | 110 | 143 |
| Total | 110 | 110 | 248 |

Laboratório Ltda., Pinhais, Paraná, 83.323–020, Brazil). Subsequently, the platelet counts were determined by using a formaldehyde in sodium citrate solution [28], a binocular microscope (Nikon E-200, Nikon Instruments Inc., Melville, New York, 11747–3064, USA.), and a Neubauer improved counting chamber (LO—Laboroptik GmbH, Friedrichsdorf, Hessen, 61381, Germany). The microhematocrit capillary tubes were centrifuged at 12,850 X G for 10 minutes and then inspected against a standard calibration to determine the packed cell volume (PCV). The samples used in the hemoglobin (Hb) analyses were prepared by pipetting 20 μl of whole blood into 5 ml of Drabkin's reagent (Newprov Produtos para Laboratório Ltda., Pinhais, Paraná, 83.323–020, Brazil) for subsequent photometric determination [28]. The erythrocyte cell indices were calculated by mean cell volume (MCV) (fl) = Hct (l) x 10/RBC ($10^{12}$/L) and by mean cell hemoglobin concentration (MCHC) (g/L) = [Hb concentration (g/L) /PCV (l)] x 100. Blood smears were treated with Wright's staining solution (Laborclin Produtos para Laboratório Ltda., Pinhais, Paraná, 83.321–210, Brazil) to conduct differential counts of 100 WBC per slide. All the parameters could not be determined for all the individuals due to lack of sample availability.

## Stress level

The stress level of each animal that was handled at the research station was estimated by assigning a score of 1–4, where 1 represents the absence of visible stress and 4 represents a high level of stress (Table 2). The cardiac rate (CR, beats/min) was recorded just after blood collection for a minimum period of 1 min. The respiratory frequency (RF, breaths/min) was monitored throughout the handling period and was recorded just after the CR was noted.

The influence of time on the blood values was evaluated by correlating the total handling time of the Amazon River dolphins–combination of specific events (Table 3)–with the hematological and physiological parameters (CR and RF). The neutrophil to lymphocyte ratio (N:L) was calculated for 24 individuals and were separated in two time categories– 18 to 31 mins of manipulation and 32 to 45 mins of manipulation–in order to study the chronology of acute stress responses. CR and RF were determined for 26 and 33 dolphins, respectively.

## Statistical analysis

Descriptive statistics were calculated for the hematological values and 90% confidence limits were used to define the reference intervals (RI) for Gaussian distributed data following the guidelines by the American Society for Veterinary Clinical Pathology for sample sizes 40 ≤ x ≤ 120 [29]. The means, standard deviations (SD), minimum and maximum values and 90% RI were estimated. Outliers were defined by the Horn's algorithm using Tukey's interquartile fences and were discarded from the analyses. Levene's test was used to assess the homogeneity

**Table 2. Four different levels of stress were attributed to Amazon River dolphins based on their behavior during handling time.**

| Level of stress | n | Behavior |
|---|---|---|
| **Stress 1 (unstressed)** | 14 | Appropriate responsiveness to external stimuli; no resistance to handling and immobilization; no vocalizations; animal is clearly relaxed. |
| **Stress 2** | 19 | Some, but not excessive, responsiveness to external stimuli; little resistance to handling and restraint; no vocalization. |
| **Stress 3** | 14 | Moderate responsiveness to external stimuli; mild resistance to handling and immobilization; occasional vocalization. |
| **Stress 4** | 19 | Extreme responsiveness to external stimuli; high resistance to handling and immobilization; frequent vocalization; temporary apnea. |

**Table 3. Specific events of the Amazon River dolphins handling during the capture-release program at RDSM in November 2004 and 2005.**

| | Small Mesh Net (min) | Boat (min) | Research Station (min) | Total Time (min) |
|---|---|---|---|---|
| Mean | 10 | 11 | 10 | 30 |
| Median | 10 | 10 | 11 | 31 |
| Min | 4 | 6 | 3 | 18 |
| Max | 16 | 16 | 16 | 45 |

of variance. The differences between the means of 2 groups (*e.g.*, males and females or pregnant and non-pregnant females) were assessed using a Student's *t*-test. Comparisons among calves, juveniles and adults, as well as among adult pregnant females, adult non-pregnant females and males, were carried out using a one-way analysis of variance (ANOVA) or Kruskal-Wallis for non-homoscedastic variances. Post hoc, Tukey's test was performed when significant differences were found. All the statistical analyses were performed using the software package *Statistica* version 7.0 (StatSoft, Tulsa, Oklahoma, 74104, USA) at a $p \leq 0.05$ level of significance.

All of the possible comparisons could not be made because of small or non-existent samples for a particular age-class or gender. As such, the sample size of each comparison may vary as it reflects the number of animals with complete data for that specific outcome. To determine the extent to which stress-related changes in hematological elements, cardiac and respiratory rates may covary with time, the strength of Pearson correlation coefficients (very weak, weak, moderate, strong, very strong) were determined [30].

## Results

One-hundred-and-ten Amazon River dolphins were successfully captured and sampled during 2 separate expeditions in November 2004 and 2005. The duration of small mesh net seining varied from 4 to 16 minutes, according to the environmental conditions and the behavior of the animals. The handling process at the research station was always performed within a predetermined safety time range at an average of 10 minutes and never exceeding 16 minutes. The combined time of handling in the small mesh net and blood collection had a mean of 30 mins and varied from 18 to 45 mins (Table 3).

### Reference ranges

Using a normal distribution, 90% double sided RI were determined after excluding the calves and the outliers through box plot and histogram analyses. The calves were not included in the RI calculation as they presented a significantly higher WBC than the adults and subadults (Table 4).

### Gender and age variation in hematology

In order to generate accurate reference values, statistical analyses were performed by separating groups according to gender, age class and reproductive status. No significant difference was found between the hematologic values of males and females. Also, no differences were observed between the pregnant ($n = 10$) and non-pregnant females ($n = 23$), pregnant females and adult males or non-pregnant females and adult males.

Since no significant differences were detected between the sexes for any parameter ($p > 0.05$), the data were pooled for age class comparisons. Significant differences in the blood

**Table 4. Hematological values for *Inia geoffrensis* captured and released at RDSM, Brazil, in November 2004 and 2005.**

| | SI Units | Adults and subadults | | | | | | | Calves | | | |
|---|---|---|---|---|---|---|---|---|---|---|---|---|
| | | n | Mean | SD | Min | Max | Reference values | | n | Mean | SD | Min | Max |
| Hct | L/L | 97 | 0,40 | 0,03 | 0,31 | 0,46 | 0,39 | 0,40 | 13 | 0,39 | 0,03 | 0,34 | 0,43 |
| RBC count | $10^{12}$/L | 92 | 3,91 | 0,65 | 2,84 | 5,93 | 3,80 | 4,02 | 13 | 3,69 | 0,43 | 2,95 | 4,55 |
| Hemoglobin | g/L | 66 | 155,97 | 31,41 | 122,00 | 390,00 | 149,52 | 162,42 | 6 | 159,13 | 14,28 | 133,00 | 175,00 |
| MCV | fL | 92 | 104 | 17 | 71 | 136 | 101 | 107 | 13 | 109 | 12 | 95 | 136 |
| MCHC | g/L | 66 | 386,8 | 36 | 298 | 469 | 379,4 | 394,1 | 6 | 400 | 49,1 | 332 | 471 |
| WBC count | $10^9$/L | 92 | 16,62* | 4,61 | 8,15 | 27,45 | 15,81 | 17,44 | 13 | 24,54* | 6,02 | 11,60 | 37,00 |
| Band neut | % | 58 | 1,06 | 1,11 | 0,00 | 4,00 | 0,83 | 1,30 | 9 | 0,89 | 1,05 | 0,00 | 3,00 |
| Band neut | $10^9$/L | 63 | 0,19 | 0,23 | 0,00 | 1,01 | 0,14 | 0,24 | 9 | 0,20 | 0,20 | 0,00 | 0,50 |
| Neutrophil | % | 58 | 48,00 | 8,30 | 32,00 | 63,00 | 46,16 | 49,65 | 9 | 45,67 | 8,83 | 35,00 | 60,00 |
| Neutrophil | $10^9$/L | 63 | 8,28 | 2,90 | 3,33 | 16,22 | 7,64 | 8,91 | 9 | 10,92 | 2,63 | 8,28 | 14,80 |
| Lymphocytes | % | 58 | 35,08 | 7,94 | 10,00 | 50,00 | 33,41 | 36,75 | 9 | 40,67 | 12,67 | 20,00 | 57,00 |
| Lymphocytes | $10^9$/L | 63 | 6,30 | 2,81 | 0,48 | 14,83 | 5,68 | 6,91 | 9 | 10,59 | 4,98 | 2,97 | 17,76 |
| Monocyte | % | 58 | 2,86 | 1,82 | 0,00 | 7,00 | 2,47 | 3,24 | 9 | 3,00 | 2,12 | 0,00 | 6,00 |
| Monocyte | $10^9$/L | 63 | 0,69 | 1,56 | 0,00 | 12,00 | 0,35 | 1,04 | 9 | 0,80 | 0,70 | 0,00 | 1,80 |
| Eosinophils | % | 58 | 13,06 | 5,73 | 4,00 | 30,00 | 11,86 | 14,27 | 9 | 9,67 | 6,42 | 0,00 | 19,00 |
| Eosinophils | $10^9$/L | 63 | 2,42 | 1,71 | 0,62 | 11,90 | 2,04 | 2,79 | 9 | 2,13 | 1,25 | 0,00 | 4,49 |
| Basophils | % | 58 | ᵃND | | | | | | 9 | ᵃND | | | |
| Basophils | $10^6$/L | 63 | ᵃND | | | | | | 9 | ᵃND | | | |
| Platelets | $10^9$/L | 87 | 297,00 | 111,95 | 78,00 | 556,00 | 277,08 | 316,99 | 13 | 296,46 | 69,57 | 194,00 | 420,00 |
| ESR | mm/hour | 88 | 50,69 | 4,95 | 20,00 | 58,00 | 49,82 | 51,57 | 10 | 48,90 | 3,45 | 44,00 | 56,00 |

[a]ND = Not determined

Significant difference with *$P < 0.001$

leukocyte counts were shown in Fig 1. Calves had higher WBC (mean = 24.54; SD = 6.02) than juveniles (mean = 18.85; SD = 5.6) (df = 102; p = 0.03) and adults (mean = 17.24; SD = 5.2) (df = 102; p = 0.0003).

Within relative WBC counts, age also affected the lymphocyte and neutrophil absolute values. The lymphocyte count was significantly higher in calves (mean = 10.59; SD = 4.98) than adults (mean = 6.20; SD = 2.70) (df = 64; p = 0.0009). In the same manner, the neutrophils count was observed to be higher in calves (mean = 10.92; SD = 2.63) than adults (mean = 8.12; SD = 2.79) (df = 64; p = 0.02) (Fig 2).

## Pregnant and non-pregnant females

The blood values of 33 adult females were assessed by ultrasound exams; 10 of these were pregnant and 23 were not. No statistically significant differences (p > 0.05) were found in any of the hematological parameters between the two groups. Although similar, this valuable information is worth noting given the scarce opportunity to examine pregnant individuals in the wild, particularly dolphins. The sample sizes, means, SD and minimum and maximum values for pregnant females are shown in the S1 Table.

## Acute capture stress influences on cardiac rate, respiratory frequency and blood values

The minimum and maximum values of CR and RF for each age class are given in Table 5. The relative stress levels were estimated for 66 individuals. Furthermore, the duration of handling

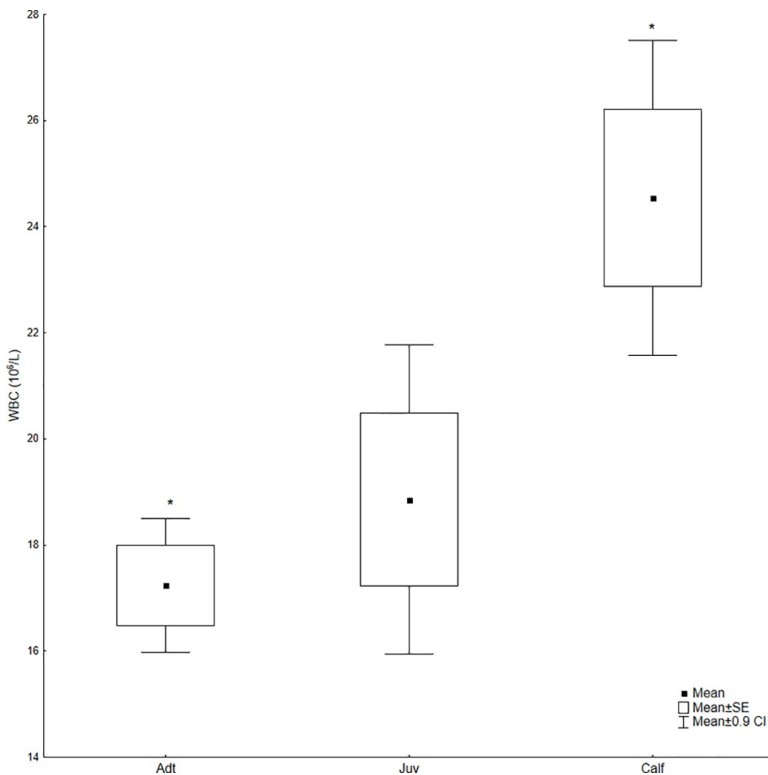

**Fig 1. WBC count for calves (n = 13), juveniles (n = 13) and adults (n = 80) of *Inia geoffrensis* from the RDSM, Amazonas, Brazil.** SE = standard error, CI = confidence interval. * Significant difference p ≤ 0.001.

time was recorded for 26 Amazon River dolphins, which was positively correlated with Hct (r = 0.61) (p = 0.002); RBC (r = 0.45) (p = 0. 04) and Hb (r = 0.54) (p = 0.01) (Fig 3).

No correlation was observed between the total handling time and estimated stress level. However, the stress level was positive and moderately correlated with the lymphocyte level (r = 0.44) (p = 0.004) and CR (r = 0.57) (p = 0.03) and positive and weakly correlated with WBC (r = 0.29) (p = 0.02), absolute neutrophil (r = 0.38) (p = 0.01), and monocyte (r = 0.34) (p = 0.03) counts, and RF (r = 0.35) (p = 0.04) (Fig 4).

Other significant correlations were noted among of the blood variables. Some of the estimates were expected to be strongly or very strongly correlated, such as neutrophil with lymphocyte percentages (r = - 0.78) (p < 0.001); WBCs with neutrophil absolute values (r = 0.85) (p = 0.000); CR and RF (r = 0.45) (p = 0.021) and Hb and MCHC (r = 0.55) (p = 0.003), which were moderately correlated since one variable was for the most part determined or influenced by the other variable. Other comparisons have revealed some more apparently dynamic interactions with moderate correlations, particularly for WBC and CR (r = 0.43) (p = 0.031), WBC and RF (r = 0.56) (p = 0.003) and neutrophils absolute values (r = 0.44) (p = 0.039) (Fig 5).

Most stressed animals (level 4) presented significantly higher lymphocyte (mean = 8.27; SD = 3.31) (df = 3; p = 0.03) counts than the animals at level 1–3 (mean = 4.99, 5.35, 7.32, respectively), whereas the least stressed animals (level 1) presented significantly less platelet counts (mean = 227; SD = 106) (df = 3; p = 0.008) than level 2 (mean = 349), level 3 (mean = 313) and level 4 animals (mean = 319).Additionally, the level of CR in the least stressed animals (mean = 67; SD = 4) (df = 3; p = 0.03) was less than those at levels 2–4 (mean = 80, mean = 86, mean = 96, respectively).

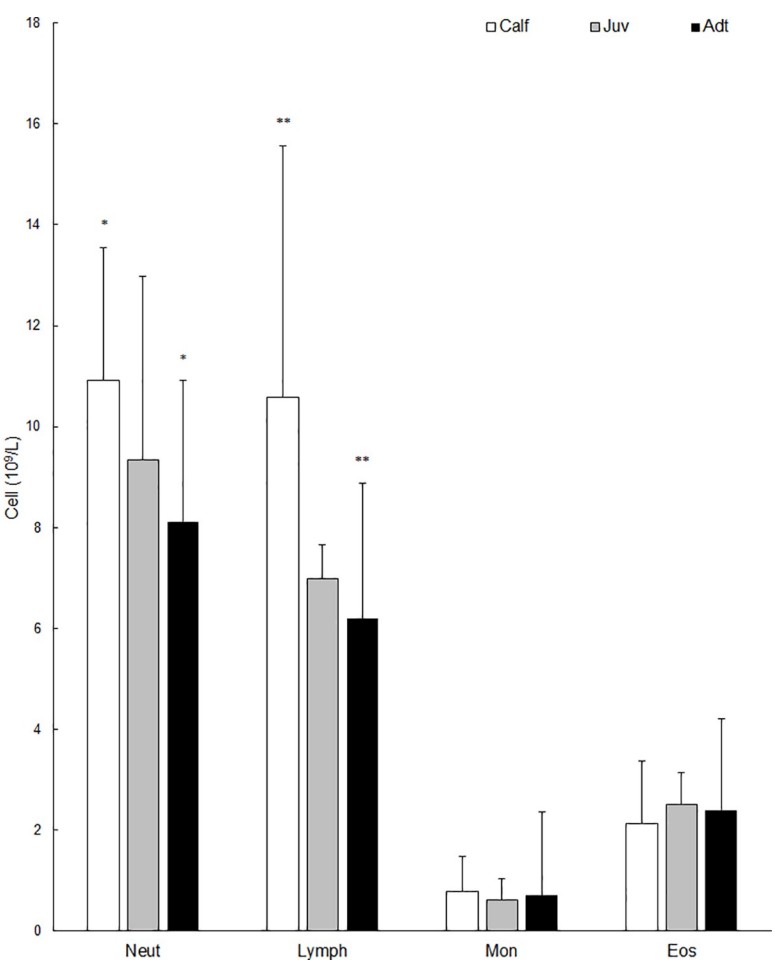

**Fig 2. Relative WBC count for juveniles (n = 13) and adults (n = 80) of *I. geoffrensis* from the RDSM, Amazonas, Brazil.** * Significant difference $p \leq 0.05$; ** Significant difference $p \leq 0.001$.

The neutrophil to lymphocyte ratio N:L was calculated with the time of capture for 24 individuals. The total time (sum of intervals 1, 2 and 3) was divided into 2 parts– 18 to 31 mins of capture and 32 to 46 mins of capture–and it was observed that the proportion of neutrophils to lymphocytes had increased from 1.4:1 to 1.7:1.

## Discussion

The goal of the estimation and use of reference values and prediction intervals for hematology data is to establish baseline values for healthy populations against which perturbations can be

**Table 5. Cardiac rate and respiratory frequency of the Amazon River dolphins during handling time at the research station in Amazonas, Brazil.**

| | Cardiac rate (beats/min) (n = 26) | | Respiratory frequency (breaths/min) (n = 33) | |
|---|---|---|---|---|
| | **Min** | **Max** | **Min** | **Max** |
| Calves | 78 | 127 | 8 | 14 |
| Juveniles | 88 | 110 | 3 | 18 |
| Adults | 60 | 113 | 3 | 18 |

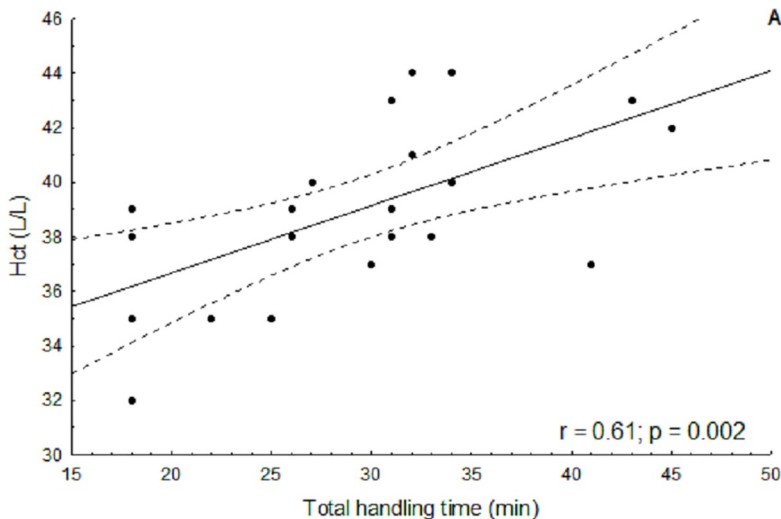

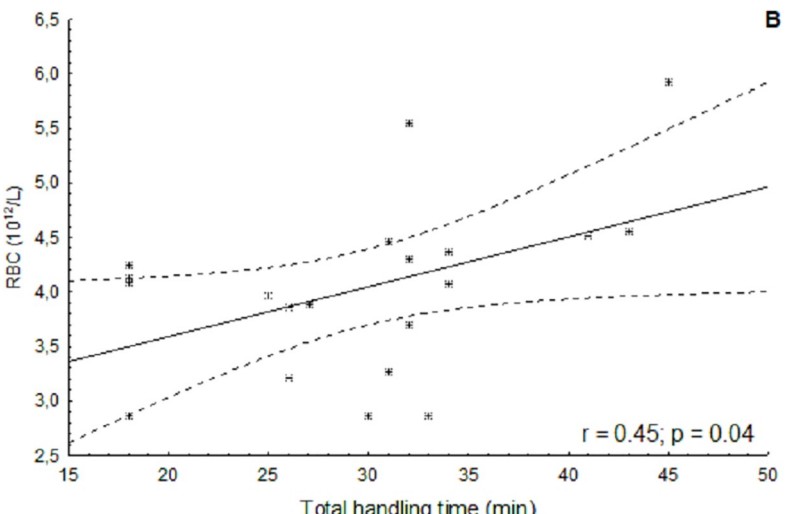

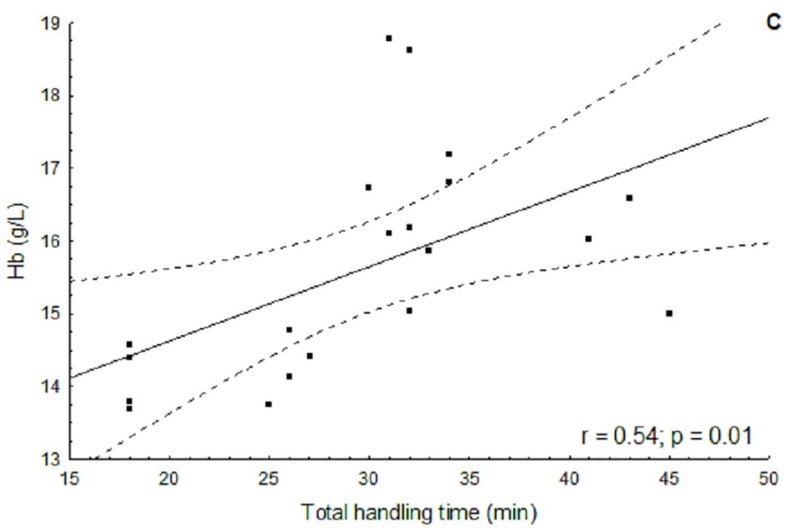

**Fig 3. Correlation of total handling time of Amazon River dolphins with blood values at RDSM in Amazonas, Brazil.** A positive correlation was found between the total handling time and Hematocrit (Hct) (A), RBC count (B) and Hb level (C).

judged in putative non-healthy populations or individuals [10]. The basis for selecting healthy populations is clearly very difficult in wild animals, but the data reported here is assumed to have been collected from healthy individuals with no obvious clinical signs of ill health. The only previously available data on the blood values of *I. geoffrensis* is from that of 22 animals that were kept in aquariums in the USA in the 1970s [31]. The Hb concentration of the free-ranging Amazon River dolphins that were examined in this study were higher (15.51 g/dl) than those of the captive individuals (13.85 g/dl) [31], which is probably a consequence of the lack of space and the shallow depth of water that these captive animals were subjected to. Also, the wild *I. geoffrensis* had higher WBC counts (18.74 x $10^3$/mm$^3$) than the captive ones (13.38 x $10^3$/mm$^3$) [31]. A major reason for the increased antigenic stimulation that was observed in the free-ranging dolphins may be due to parasitic infections, which consequently increase eosinophil counts [32]. Although it has not been documented yet, the probable higher degree of parasitism experienced by wild Amazon River dolphins as compared to the captive ones may lead them to have a higher percentage of eosinophil (5% and 13%, respectively). Besides, the captive dolphins were also less likely to be stressed than the wild dolphins, which has also affected the WBC count. Subclinical bacterial, fungal, protozoal and viral infections should also be considered as factors for elevated WBC in wild populations as compared to the managed populations [33].

Erythrocyte sedimentation rate (ESR) is a nonspecific indicator of inflammation acute phase response and is used as a monitoring tool in veterinary and human medicine [3]. Although it is not commonly cited in studies conducted on the blood values of marine mammals, the significance and application of ESR measurements in dolphins is likely to be

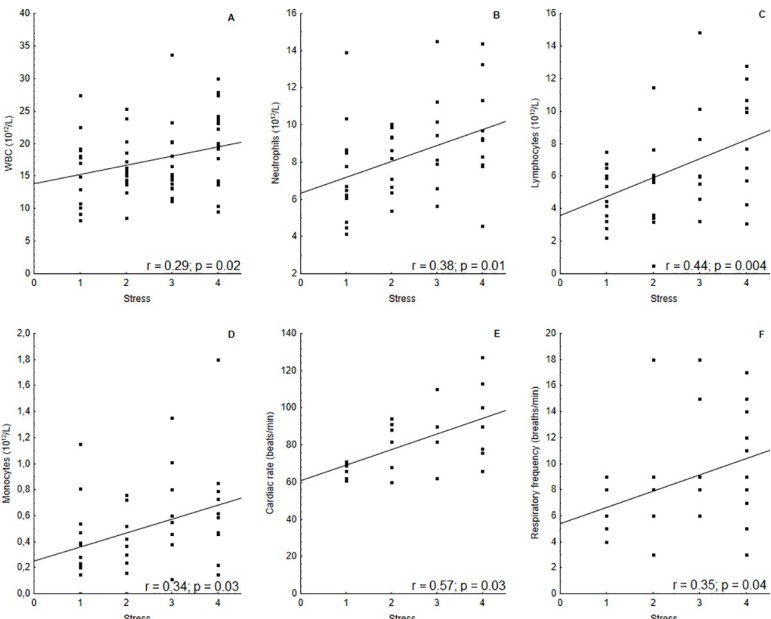

**Fig 4. Correlation of stress level with blood and physiological parameters of captured-released *I geoffrensis* at RDSM in Amazonas, Brazil.** The stress level was positively correlated with WBC (A), neutrophils (B), lymphocytes (C), monocytes (D), cardiac rate (E) and respiratory frequency (F).

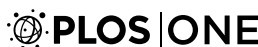

**Fig 5. Correlation of blood values and physiological parameters of captured-released *I geoffrensis* at RDSM, Amazonas, Brazil.** Positive correlation was found between WBC with CR (A) and RF (beats/min) (B) and neutrophils and RF (C).

correlated with diseases and other sources of inflammation and may be a useful prognostic indicator [3]. While ESR values are very species-specific and tend to vary considerably among marine mammals (*e. g.* 16–52 mm/hour for short-finned pilot whale *Globicephala macrorhynchus*; 3–29 mm/hour for false killer whale *Pseudorca crassidens*) [34], the CI 90% found in the 102 animals that were sampled in the present study (50–51 mm/h) indicated a very narrow range, which may represent a reliable parameter to monitor infections in river dolphins, as all the animals included in the analysis were apparently healthy. However, no validation was done on the animals with inflammation and the response to it regarding the ESR.

While the blood values were similar among age classes, some differences were noted. Calves presented higher WBC counts, which was followed by juveniles and adults. Higher leukocyte indices may be due to the developing immune systems of the younger animals as they mount immunologic responses to a broader range of largely novel antigenic stimuli [35]. The influence of age on blood values in aquatic mammals has been documented in previous studies [3,9,36,37]. Older animals tended to have lower WBC counts than juveniles and calves of the false killer whale (*Orcinus orca*), bottlenose dolphin and beluga whale (*Delphinapterus leucas*) [16,35]. In the WBC differential count, calves had a higher absolute value of lymphocytes as compared to juveniles and adults, while neutrophils were more pronounced in adults.

Consistent studies of blood values in wild pregnant female dolphins are very scarce, which is mostly because of the major logistical difficulties in obtaining samples from free-ranging cetaceans. Here, the blood values of 33 adult females that had their reproductive status assessed by an ultrasound exam showed no difference between pregnant and nonpregnant animals. Similar results were reported on bottlenose dolphins from the Indian River Lagoon, Florida, USA [9]. Alternatively, pregnant bottlenose dolphins from Charleston, South Carolina, USA, had statistically lower Hct as compared to both the non-pregnant females and males, and the authors attributed this observation to the physiologic hemodilution caused by pregnancy [3]. The physiological demands of the fetus and pregnancy itself on the mother varies according to the stage of pregnancy in cetaceans [38]. As they were free ranging animals, the sampled female *I. geoffrensis* in this study presented varied pregnancy stages, which can also explain the lack of blood value differences between pregnant and non-pregnant females.

Handling and transportation are known to be stressful for wild terrestrial mammals, and it is likely that the same holds true in aquatic mammals; however, few studies have reported this occurrence in dolphins [39]. A significant challenge of studying stress in marine mammals, or any wild species for that matter, is obtaining the baseline data that represents an unstressed state of being. Pursuit, capture, restraint and sampling procedures are obvious stressors that can influence analyses, sometimes within minutes [34]. Projeto Boto [25], a long-term project that involved the capture and release of freshwater dolphins in the Amazon region of Brazil, granted us a very good opportunity to take the first steps in understanding the physiological changes in the blood parameters of Amazon River dolphins under different degrees of stress.

Induced alterations in blood values driven by stressors can be due to different stimuli in mammals' bodies. Leukocytosis and neutrophilia can be induced by epinephrine mobilization of these cells from the bone marrow [40]. The handling of the dolphins here as a potential source of stress, showed a positive correlation in the duration of handling to the RBC, Hb concentration and Hct. Similar results of higher Hb and platelet counts were observed in pantropical spotted dolphins (*Stenella attenuata*) that were under stress during an encirclement by a

tuna purse seine operation in Mexican waters [8]. Increase in RBC counts, Hb concentration and Hct were associated with the stress-mediated decrease in calculated plasma in women under acute stress. The plasma volume decreased secondarily to a rise of circulating total cholesterol, high density lipoprotein cholesterol, low density lipoprotein cholesterol, and triglyceride concentrations [41]. Further investigation on these plasma elements would be necessary to confirm the cause of RBC, Hb and Hct rise in dolphins under stress.

Besides blood value alterations, acute stress conditions can be visually detected through behavioral alterations. Here, the WBC, absolute counts of neutrophils, lymphocytes and monocytes showed a positive correlation with the stress level, as well as with the CR and RF. This indicates that the empirical observations of stress and its classification/hierarchy based on the degree of stress can be used to assess alterations in the blood cell values and physiological response to it. It is likely that WBC may increase under stressful conditions, as it was documented in the case of pantropical spotted dolphins in the southern coast of Mexico where the time of being chased and captured was correlated with an increase in WBC during tuna fishing [8].

The positive correlation between the RF and WBC is probably due to a more pronounced increase of neutrophils than other types of leucocytes during stress. The increase in the proportion of N:L reinforces the finding that the occurrence of a neutrophil increases during acute stress. However, it should also be pointed out that although both observations could be considered indicative of acute stress, the relationships found here could also represent a normal physiological state than a signal of stress, especially for immature dolphins. Younger animals can have a naturally higher percentage of WBCs due to their developing immune systems, while their RF tends to be higher because of a normally elevated metabolism [42]. This combination may have led to some error in the correlation analysis between RF and WBC.

Given the very clear effect of stress related glucocorticoid hormones on leukocyte profiles, an increase in the N:L ratios is an expected observation of stress response. The N:L ratio has been shown to increase in a variety of mammals following transportation, including horses, goats, swine and cattle [43], as well as the Amazonian manatee (*Trichechus inunguis*) [44], dolphins [39] and black rhinoceros (*Diceros bicornis*) [45]. "Typical stress leukogram" was observed during a mass stranding of 17 striped dolphins (*Stenella coeruleoalba*) along with mild relative neutrophilia and absolute lymphopenia and eosinopenia [46]. Despite the low number of animals in this specific analysis, a similar increase in the N:L occurred in *I. geoffrensis* in proportion to the duration of capture and handling. Also, the relative white blood cell counts appear to be an efficient method of assessing acute stress in free-ranging river dolphins.

## Conclusions

The baseline values for the hematological parameters of an apparently healthy river dolphin population of Amazon River dolphin (*I. geoffrensis*) have been described for the first described. Calves had higher neutrophil and lymphocyte absolute counts, which was reflected in a higher overall WBC count. Empirical observations of stress based on its intensity/degree can be used to assess alterations in the blood cell values and physiological response to it. Variations in Hct, Hb, N:L ratio, WBC, platelet counts, neutrophils, lymphocytes and monocytes absolute counts proved to be a good bioindicator of acute stress in Amazon River dolphins. CR and RF also showed a positive correlation to the level of stress in these animals. The baseline values are of crucial importance to efficiently monitor the health of wild populations, and the data provided here represents a major contribution on this matter. In addition, the hematological blood parameters of such populations from an area with low human impact could serve as a trustworthy baseline for other dolphin populations in the Amazon region.

## Supporting information

**S1 Table. Hematological values for adult female *Inia geoffrensis*.**
(XLSX)

## Acknowledgments

The authors are very grateful to all the people involved in the capture, restraint and blood collection of the animals used in this work, especially Dr. Anthony Martin, who provided the necessary support for this study. We thank Rodrigo Amaral for revising the manuscript. This study was a part of Projeto Boto, a cooperative agreement between the National Institute for Amazonian Research–INPA/MCTI–and the Mamirauá Sustainable Development Institute–MSDI-OS/MCTI.

## Author Contributions

**Conceptualization:** Daniela M. D. de Mello.

**Data curation:** Daniela M. D. de Mello.

**Formal analysis:** Daniela M. D. de Mello.

**Funding acquisition:** Vera M. F. da Silva.

**Investigation:** Vera M. F. da Silva.

**Methodology:** Daniela M. D. de Mello.

**Project administration:** Vera M. F. da Silva.

**Supervision:** Vera M. F. da Silva.

**Writing – original draft:** Daniela M. D. de Mello, Vera M. F. da Silva.

**Writing – review & editing:** Daniela M. D. de Mello, Vera M. F. da Silva.

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
