## [Decision Letter · Decision Letter 0]

6 Nov 2019

PONE-D-19-17847

Hematologic profile of Amazon river dolphins Inia geoffrensis and its variation during acute capture stress

PLOS ONE

Dear Dr. Magalhães Drummond de Mello

Thank you for submitting your manuscript to PLOS ONE. After careful consideration, we feel that it has merit but does not fully meet PLOS ONE’s publication criteria as it currently stands. Therefore, we invite you to submit a revised version of the manuscript that addresses the points raised during the review process.

Both reviews think this manuscript is an important contribution to the veterinary field, particularly since it increases knowledge on a rare an vulnerable species, the Amazon river dolphin.  However, as you will see, one reviewer wants more clarity regarding which individuals and in which way you conducted the statistical analyses.  Both reviewers also suggest you to have a native english speaker helping you with some of the grammar of the manuscript.

I also want to apologize for the long time since you submitted the manuscript until you received a decision.  It was very hard to find the right reviewers for this manuscript, with more than 20 potential reviewers rejecting to review due to the specific topic of the manuscript.

We would appreciate receiving your revised manuscript by 1st of December.  To enhance the reproducibility of your results, we recommend that if applicable you deposit your laboratory protocols in protocols.io, where a protocol can be assigned its own identifier (DOI) such that it can be cited independently in the future. For instructions see: http://journals.plos.org/plosone/s/submission-guidelines#loc-laboratory-protocols

We look forward to receiving your revised manuscript.

Kind regards,

Susana Caballero, PhD

Academic Editor

PLOS ONE

Journal Requirements:

3.  We note that [Figure 1] in your submission contains a  map  image which may be copyrighted. All PLOS content is published under the Creative Commons Attribution License (CC BY 4.0), which means that the manuscript, images, and Supporting Information files will be freely available online, and any third party is permitted to access, download, copy, distribute, and use these materials in any way, even commercially, with proper attribution. For these reasons, we cannot publish previously copyrighted maps or satellite images created using proprietary data, such as Google software (Google Maps, Street View, and Earth). For more information, see our copyright guidelines: http://journals.plos.org/plosone/s/licenses-and-copyright.

You may seek permission from the original copyright holder of Figure(s) [1] to publish the content specifically under the CC BY 4.0 license. 

If you are unable to obtain permission from the original copyright holder to publish these figures under the CC BY 4.0 license or if the copyright holder’s requirements are incompatible with the CC BY 4.0 license, please either i) remove the figure or ii) supply a replacement figure that complies with the CC BY 4.0 license. Please check copyright information on all replacement figures and update the figure caption with source information. If applicable, please specify in the figure caption text when a figure is similar but not identical to the original image and is therefore for illustrative purposes only.

3. Please include a new copy of Table 4 in your manuscript; the current table is difficult to read. Please follow the link for more information: http://blogs.PLOS.org/everyone/2011/05/10/how-to-check-your-manuscript-image-quality-in-editorial-manager/

Additional Editor Comments (if provided):

See above

Reviewers' comments:

Reviewer's Responses to Questions

**Comments to the Author**

1. Is the manuscript technically sound, and do the data support the conclusions?

Reviewer #1: Partly

Reviewer #2: Yes

2. Has the statistical analysis been performed appropriately and rigorously? 

Reviewer #1: I Don't Know

Reviewer #2: Yes

3. Have the authors made all data underlying the findings in their manuscript fully available?

Reviewer #1: Yes

Reviewer #2: Yes

4. Is the manuscript presented in an intelligible fashion and written in standard English?

Reviewer #1: No

Reviewer #2: Yes

5. Review Comments to the Author

Reviewer #1: Overall Comment:

The data presented here represent important contributions to the literature regarding blood reference ranges for the understudied boto population. I do have some reservations regarding its publication at this time, and request a revision for the following reasons.

I am concerned about the inclusion of animals with high stress scores in the creation of baseline values/reference ranges. Ideally, animals with high stress scores would be excluded from the reference range creation since they are not “normal.” I see this paper as having two separate purposes, 1) to establish hematological reference ranges for minimally stressed boto by age class and 2) to evaluate the effect of stress on hematological parameters in this species. I think it would be a stronger manuscript if organized as such. If you choose not to split the study out in this manner, there should be some mention of this as a study limitation in the discussion (the inclusion of stressed animals when establishing baselines).

I was unclear whether or not all the data were tested for a normal or Gaussian distribution. Especially for some of the parameters with relatively low sample sizes, this is extremely important with regards to the question on whether rigorous statistical methods were applied. I answered “I don’t know” because this is a pending question I have for the authors in my review.

While the authors in general did a good job of translating to English, the language is still awkward in many places and there are still some confusing statements and grammatical errors that appear to be associated with translation. I tried to assist and interpret where I could, but I recommend a native English speaker edit the manuscript prior to resubmission.

Abstract:

Line 30, 36: Change “gender” to “sex” throughout text as sex refers to the biological differences between male and female, and gender refers to the individual’s identity and is therefore not suitable for use with animals.

Lines 36-40: Since reference ranges are often created according to sex, reproductive status, and age class, I would recommend moving these findings up to line 33 before the sentence starting “Means…” You have essentially created age-class specific reference ranges in this study since those were the groups with statistically significant differences – and I would clarify that in the abstract.

Lines 38-39: I recommend deleting “as a result of the still-developing immune system.” There are other potential explanations for calves having higher WBCs (higher lung worm burden in younger animals - which has been observed in other dolphin populations, or higher stress level than older animals). Given these alternate possible explanations and the fact that this is not a common finding in all populations of dolphins, I do not recommend being so definitive in explaining its cause.

Introduction:

Line 52: Change “accessed” to “assessed”

Line 55: Recommend changing “health status approaches” to “health assessments”

Line 56-57: All these studies can be referred to as “capture-release” studies, I am unclear why the distinction is being made for the St. Aubin study as “netted wild population” vs capture release programs. Please clarify.

Line 57: Change semicolon to comma and “parameters” to “parameter” and “baseline” to “baselines”

Lines 58-62: This sentence is confusing the way it is written, please reword. With reference #11 – there had not been a baseline study prior to the oil spill on that particular population (although baseline studies existed for other Tt populations that were used for comparison). I would be more explicit in explaining how this particular citation makes the authors’ point. I also think the reference to seasonal variation in the Sarasota dolphins is confusing since you do not address seasonality at all in the remainder of the paper (even though it is controlled for with your sample timing).

Line 64: delete “aged”

Line 65: This is not necessarily a common finding among the capture-release sampled dolphins in various locations. Given the age of this citation, I would recommend either striking this comment or finding additional more recent citations to make a stronger case.

Line 67: Change “alteration on” to alterations in”

Line 69: change “englobe” to “involve”

Line 70: change “may reflect” to “may be reflected in”

Line 71: change “gender” to “sex”

Line 73: I believe you mean “before establishing baseline blood parameters.” Ideally you are establishing these reference ranges with animals that are minimally stressed and removing the outliers from your “normal” sampling population.

Line 76: include citation for the most recent IUCN report

Line 76-77: Recommend changing “under intentional catch” (which is vague) to “actively fished commercially”

Line 78: Recommend changing “leading to significant populations decrease” to “leading to a significant population decline”

Lines 83-84: delete comma after “reserve” and replace with “in the”

Line 87: Why single out the ESR here? Seems unnecessary – either please justify or delete.

Materials and Methods:

Line 109: Recommend spelling out the RDSM acronym in the figure caption

Line 116: Change “proximities of research floating station” to “proximity of the floating research station”

Lines 116-117: Recommend changing the sentence starting with “It varied” to: “The number and behavior of encircled animals varied according to environmental conditions.”

Line 118: If you are going to indicate the depth of the channel it would also be helpful to know the depth of your capture net (s).

Lines 119-121: You state that the usual method was to seine the dolphins to the shoreline. That leads me to believe that there were other methods used to capture the dolphins in some instances. If so, please describe those methods as well, if not, please clarify the text here.

Line 126: What were your criteria for determining pregnancy? Identification of a fetus? Recommend explicitly stating that.

Line 139: Did you actually use a catheter or a needle? Was the blood drawn directly into the EDTA vacutainer tube or was it drawn into a syringe and then transferred to the tubes? The way this is written, it sounds like the blood from the EDTA tubes was transferred to a tube with no anticoagulant and allowed to clot, which is confusing. Please clarify your methods.

Line 146: How were these manual methods standardized? Were they consistently done by a single trained individual or were multiple individuals trained on standardized methods?

Line 171: For stress levels 1 & 2 behavior, perhaps you mean no/little excessive response to external stimuli? Or appropriate response to external stimuli? Relaxed dolphins should be alert and responsive to external stimuli, but not hyper-responsive. By immobilization do you mean restraint? Recommend re-wording these behavioral descriptions.

Line 175: I think you mean “the neutrophil to lymphocyte ratio” ?

Lines 176-177: 12 individuals in each category? Are samples from the same individual compared between the two time categories or are the comparisons done between different animals? Please clarify the methods here.

Line 177: insert ‘of’ between ‘chronology’ and ‘acute’

Line 181: Recommend first discussing the tests for significant differences in age class, reproductive status, and sex, then discuss the methods for creating reference ranges based on those outcomes. Were all the data tested for normality (Gaussian distribution)? Especially with some of the parameters that had smaller sample size, normal distribution cannot be assumed and should be tested.

Line 200: Strength of correlation (weak, moderate, or strong according to r value) should be stated here. You can find references in the literature to describe correlation strength.

Line 195: How were multiple comparisons controlled for in order to prevent Type I error (false positives)? You compared a lot of variables and should either control for these multiple comparisons or justify why you did not do so.

Line 214: Since Table 4 is cut off, I cannot see if you stated 90% RI for the calves. I am assuming you did not due to small sample size. I would explicitly state what you are presenting for the calf data here and why there are not reference ranges established. I would also recommend removing the moderate and severely stressed dolphins from the reference range calculation.

Line 220: Formatting in the version I have cuts off the left (parameter) and right (last column is calves min) margins of this table. Cannot fully evaluate data presented. Recommend including more details regarding the reference values in the table caption (90% double sided reference intervals)

Line 221: In the methods, it stated the level of significance was set at p < 0.05 but here it states p < 0.001. Why the discrepancy?

Line 234: In Fig 2: Recommend adding sample sizes (n) above each age class box plot

Line 242: Again, recommend including sample sizes in Fig 3 (either in the caption or in the legend)

Line 251: I do not see medians in table S1, delete reference to them here.

Line 252: These are not reference intervals, recommend deleting ”RI” and just referring to the data in table S1 as summary statistics.

Line 263: Fig 4: Recommend including r and p values either on the figure or in the caption. Same comment for Figs 5 and 6 as well.

Line 254: Recommend spelling out cardiac rate and respiratory frequency in this section title as these are not common abbreviations.

Line 268: Correlation should be categorized as weak, moderate or strong based on the r values.

Discussion:

Line 300: change “comes” to “come” – “data” is a pleural noun (singular “datum”)

Line 301: change to “in the 1970’s”

Lines 304-310: Captive boto were also likely less stressed than the wild-caught boto, which would also affect the WBC count. Subclinical bacterial, fungal, protozoal, and viral infections should also be included in the differentials for elevated WBC in wild vs managed populations.

Line 320: change “an” to “a” Also, without demonstrating a correlation between ESR and inflammation, I would caution against evaluating this parameters utility here. While the narrow range MAY indicate that this could be a useful indicator IF it does in fact change with inflammation, you have not proven the latter yet.

Lines 318-320: Was there any correlation between ESR and WBC count? If not, this makes it more likely that the WBC/neut elevations were due to stress rather than inflammation/infection/antigenic stimulation.

Lines 321-330: Also need to include stress as a differential for higher WBC counts in juveniles. And a note that this is not a consistent finding among all cetacean populations.

Line 335: change “was” to “were”

Line 340: Change “towards” to “on”

Line 342: Can you present data in the results regarding the sample sizes according to each trimester if available to help elucidate this point?

Line 343: change “values” to “value” and “towards non-pregnant females” to “between pregnant and non-pregnant females”

Lines 356-358: Due to the spleen’s small size relative to the dolphin’s overall body mass, my understanding is that unlike in other species, splenic contraction is not considered a significant differential for elevated RBC, HCT, Hb in cetaceans.

Line 359: Recommend using the term “epinephrine” rather than “adrenalin”

Line 369: change “access” to “assess”

Line 373-374: Also cannot rule out subclinical pulmonary disease as contributing to this trend, and this should be included as a differential for that finding.

Lines 376-378: Please reword this sentence for clarity.

Line 379: Add “can” between ‘animals’ and ‘have’ – as this is not always a consistent finding in cetaceans

Lines 384-394: More clarification in your discussion of the different types of stress leukograms is warranted – since you have the expected neutrophilia with a corticosteroid stress response, but not the associated lymphopenia (as with the “typical stress leukogram” that you cite). Lymphocytosis would be associated with an epinephrine-induced response. While you mention both of these hormone-induced pathways, your explanation of the lymphocytosis is incomplete.

Line 400: change ‘access’ to ‘assess’

Line 403: change “ad” to “and”

Reviewer #2: I attach minor corrections per line.

The title identifies the study subject as Amazon river dolphin, also known as boto. But the interchange of this common name in the manuscript may be confusing to the reader. I would recommend being consistent in using one. Mention both names once, but after that be consistent in using one.

In Spanish and perhaps in Portuguese, the names of rivers begin with capital letters, but the word river itself does not begin with capital letters, i.e., río Amazonas. But in English, both the river name and the word river begin with capital letters, i.e., Amazon River. Thus, in the title and throughout the document, it should be Amazon River dolphin.

All animals, except common animals (horses, goats, swine, cattle), should be first introduced with a common name followed by its scientific name. Afterwards, I recommend using just the common name.

Add the common names for Globicephala macrorhynchus, Pseudorca crassidens, Orcinus orca, Delphinapterus leucas, Stenella coeruleoalba

Add the scientific names for Amazonian manatee, rhinoceros

31: 110 write out as One-hundred-and-ten

38: Delete The before calves

40: should be lymphocytes

42: should be neutrophils, lymphocytes and monocytes

44: define Hct and then put in parenthesis. Red Blood Cells should be red blood cells. Define Hb and then put in parenthesis.

65: should be white blood cells (WBC)

66: should be hematocrit (Hct)

74: should be The Amazon river dolphin or boto (Inia geoffrensis)

76: cite where the IUCN lists the species as endangered.

76: instead of “under intentional catch” use affected by intentional capture

81: delete or boto (I. geoffrensis)

83: should be Mamirauá Sustainable Development Reserve (RDSM) and Amazon River.

103: delete Mamirauá Sustainable Development Reserve and delete the parenthesis in RDSM

144: Comment: the red top tubes were allowed to clot at ambient temperature, room temperature or in the refrigerator? Please specify.

147: delete white blood cell and delete delete the parenthesis in WBC

167: should be Cardiac rate (CR, beats/minute)

168: should be respiratory frequency (RF, breaths/minute)

174: Should be (Table 3)

178: should be CR and RF were

202: 110 write out as One-hundred-and-ten

202-203: Delete by Projeto Boto,

214-217: box plot and histogram analyses (Table 4). Then delete line 215, 216, 217.

231: change white blood cells counts for WBC

234: Delete Blood cell (and the other ).

270: cardiac rate should be CR, and respiratory frequency should be RF

277: understandably/expected should be understandably and expected

279: cadiac rate and respiratory frequency should be CR and RF

280: put a comma (,) after (p=0.003)

282: respiratory frequency should be RF

283: respiratory frequency should be RF

287: respiratory frequency should be RF

288: cadiac rate should be CR

301: by the 1970s. It does not 1970’s because it is not possessive. It is plural, so it should be 1970s

305: Where it says I. geoffrensis, change for ones

317: use common name before Globicephala macrorhynchus and Pseudorca crassidens

327: use common names for Orcinus orca, T. truncatus, Delphinapturus leucas in conjunction with scientific name

356: Hb instead of hemoglobin

362: the scientific name for the pantropical spotted dolphin is Stenella atteanuata. Stenella tropicalis does not exist.

373: should be RF for respiratory frequency

375, 385, 386, 391: Explain first what is N:L before using it.

387: follow the scientific name for Amazonian manatee

388: rhinos is a popular way to refer to rhinoceros. However, which species of rhinoceros are you referring to? White rhinoceros (Ceratotherium simum), black rhinoceros (Diceros bicornis), Indian rhinoceros (Rhinoceros unicornis), Javan rhinoceros (Rhinoceros sondaicus) or Sumatran rhinoceros (Dicerorhinus sumatrensis). Or is it to species of the Rhinocerotidae?

389: common name for Stenella coeruleoalba

397: instead of boto, use Amazon River dolphin

401: use Hct and Hb

403: use CR and RF. The and is missing an n.

Figure 1: Legends inside figure should be in English or the same terms used in the text.

Very good clinical work, and much needed for this species.

Congratulations in this accomplishment.

6. PLOS authors have the option to publish the peer review history of their article (what does this mean?). If published, this will include your full peer review and any attached files.

Reviewer #1: No

Reviewer #2: Yes: Antonio A. Mignucci-Giannoni

---

## [Author Response · Author response to Decision Letter 0]

29 Nov 2019

Dear Reviewers,

Please find my responses to all of your corrections, suggestions and comments. I believe the modifications suggested by you were every important to enhance the manuscript quality. Despite all comments have been answered, I would like to point out some modifications. The manuscript has been reviewed by a native English speaker. Figure 1 was removed from the text and some information was included on figures 3-5. The permit number was included. 

Again, I would like to thank for all your valuable suggestions. 

Best regards,

Daniela.

PONE-D-19-17847

Hematological profile of Amazon river dolphins Inia geoffrensis and its variation during acute capture stress. 

de Mello & da Silva

• The title identifies the study subject as Amazon river dolphin, also known as boto. But the interchange of this common name in the manuscript may be confusing to the reader. I would recommend being consistent in using one. Mention both names once, but after that be consistent in using one.

A: The term “boto” was replaced by Amazon river dolphins throughout the text, with exception of “Projeto Boto” 

• In Spanish and perhaps in Portuguese, the names of rivers are begin with capital letters, but the word river itself does not begin with capital letters, i.e., río Amazonas. But in English, both the river name and the word river begin with capital letters. Thus, in the title and throughout the document, it should be Amazon River dolphin. 

A: Thank you for your observation, the word river was modified throughout the manuscript.

• All animals, except common animals (horses, goats, swine, cattle), should be first introduced with a common name followed by its scientific name. Afterwards, I recommend using just the common name. 

• Add the common names for Globicephala macrorhynchus, Pseudorca crassidens, Orcinus orca, Delphinapterus leucas, Stenella coeruleoalba

A: Attended

• Add the scientific names for Amazonian manatee, rhinoceros

• A: Attended

Lines:

31: 110 write out as One-hundred-and-ten

A: Attended

38: Delete The before calves

A: Attended

40: should be lymphocytes

A: Attended

42: should be neutrophils, lymphocytes and monocytes

A: Attended

44: define Hct and then put in parenthesis. Red Blood Cells should be red blood cells. Define Hb and then put in parenthesis.

A: Attended

65: should be white blood cells (WBC)

A: Attended

66: should be hematocrit (Hct)

A: Attended

74: should be The Amazon river dolphin or boto (Inia geoffrensis)

A: Attended

76: cite where the IUCN lists the species as endangered.

A: Attended

76: instead of “under intentional catch” use affected by intentional capture

A: Attended

81: delete or boto (I. geoffrensis)

A: Attended

83: should be Mamirauá Sustainable Development Reserve (RDSM) and Amazon River.

A: Attended

103: delete Mamirauá Sustainable Development Reserve and delete the parenthesis in RDSM

A: Attended

144: Comment: the red top tubes were allowed to clot at ambient temperature, room temperature or in the refrigerator? Please specify.

A: Ambient temperature, information included.

147: delete white blood cell and delete delete the parenthesis in WBC

A: Attended

167: should be Cardiac rate (CR, beats/minute)

A: Attended

168: should be respiratory frequency (RF, breaths/minute) 

A: Attended

174: Should be (Table 3)

A: Attended

178: should be CR and RF were

A: Attended

202: 110 write out as One-hundred-and-ten

A: Attended

202-203: Delete by Projeto Boto,

A: Attended

214-217: box plot and histogram analyses (Table 4). Then delete line 215, 216, 217. 

A: Attended

231: change white blood cells counts for WBC

A: Attended

234: Delete Blood cell (and the other ).

A: Attended

270: cardiac rate should be CR, and respiratory frequency should be RF

A: Attended

277: understandably/expected should be understandably and expected

A: Attended

279: cadiac rate and respiratory frequency should be CR and RF

A: Attended

280: put a comma (,) after (p=0.003)

A: Attended

282: respiratory frequency should be RF

A: Attended

283: respiratory frequency should be RF

A: Attended

287: respiratory frequency should be RF

A: Attended

288: cadiac rate should be CR

A: Attended

301: by the 1970s. It does not 1970’s because it is not possessive. It is plural, so it should be 1970s

A: Attended

305: Where it says I. geoffrensis, change for ones

A: Attended

317: use common name before Globicephala macrorhynchus and Pseudorca crassidens

A: Attended

327: use common names for Orcinus orca, T. truncatus, Delphinapturus leucas in conjunction with scientific name

A: Attended

356: Hb instead of hemoglobin

A: Attended

362: the scientific name for the pantropical spotted dolphin is Stenella atteanuata. Stenella tropicalis does not exist. 

A: Thanks for your observation. “S. tropicalis” was funny…

373: should be RF for respiratory frequency

A: Attended

375, 385, 386, 391: Explain first what is N:L before using it. 

A: Information added to the material and methods section. 

387: follow the scientific name for Amazonian manatee

A: Attended

388: rhinos is a popular way to refer to rhinoceros. However, which species of rhinoceros are you referring to? White rhinoceros (Ceratotherium simum), black rhinoceros (Diceros bicornis), Indian rhinoceros (Rhinoceros unicornis), Javan rhinoceros (Rhinoceros sondaicus) or Sumatran rhinoceros (Dicerorhinus sumatrensis). Or is it to species of the Rhinocerotidae?

A: Black rhinoceros, information included

389: common name for Stenella coeruleoalba

A: information included

397: instead of boto, use Amazon River dolphin

A: Attended

401: use Hct and Hb

A: Attended

403: use CR and RF. The and is missing an n.

A: Attended

Figure 1: Legends inside figure should be in English or the same terms used in the text. 

A: Figure 1 was removed from the manuscript.

Very good clinical work, and much needed for this species. 

Congratulations in this accomplishment. 

Thank you :D

Review of “Hematologic profile of Amazon river dolphins Inia geoffrensis and its variation during acute capture stress”

Overall Comment: 

The data presented here represent important contributions to the literature regarding blood reference ranges for the understudied boto population. I do have some reservations regarding its publication at this time, and request a revision for the following reasons.

I am concerned about the inclusion of animals with high stress scores in the creation of baseline values/reference ranges. Ideally, animals with high stress scores would be excluded from the reference range creation since they are not “normal.” I see this paper as having two separate purposes, 1) to establish hematological reference ranges for minimally stressed boto by age class and 2) to evaluate the effect of stress on hematological parameters in this species. I think it would be a stronger manuscript if organized as such. If you choose not to split the study out in this manner, there should be some mention of this as a study limitation in the discussion (the inclusion of stressed animals when establishing baselines).

A: Yes, the study had two main goals: first to establish reference blood values for the species e second to see with there is any variation on these values regarding age, sex, stress level… I understand your point, but still I believe all animals (with exception to outliers) should be included in the reference ranges since there is no other way to capture and restraint wild animals without causing any stress. As we have included values from both situations - less and more stressed animals – we have created a range of blood values that represent healthy individuals under different degrees of stress. This will be better explained in the discussion. Also, I have included in the manuscript comparisons of blood values among different stress level. 

I was unclear whether or not all the data were tested for a normal or Gaussian distribution. Especially for some of the parameters with relatively low sample sizes, this is extremely important with regards to the question on whether rigorous statistical methods were applied. I answered “I don’t know” because this is a pending question I have for the authors in my review. 

A: I understand your concern about the normality of the data. Same occurred to one reviewer when I have presented my masters study (same as this one), I had to show him my raw data and all analyses I had performed. If there is any particular data you want to have a look, I am happy to send you. Or if there are any specific values that you believe I should test again, please let me know. Besides the relatively low number of individuals, the variation of the blood values was not very pronounced, I believe that’s the reason the distribution was normal and the variances were homocedastic for most parameters and comparisons. The normality was first observed by histogram graphs, outliers were identified and excluded from the analysis which promoted a subsequent adjustment of the Gaussian curve. 

While the authors in general did a good job of translating to English, the language is still awkward in many places and there are still some confusing statements and grammatical errors that appear to be associated with translation. I tried to assist and interpret where I could, but I recommend a native English speaker edit the manuscript prior to resubmission.

A: The manuscript has been reviewed by a native English speaker.

Abstract:

Line 30, 36: Change “gender” to “sex” throughout text as sex refers to the biological differences between male and female, and gender refers to the individual’s identity and is therefore not suitable for use with animals. 

A: Attended

Lines 36-40: Since reference ranges are often created according to sex, reproductive status, and age class, I would recommend moving these findings up to line 33 before the sentence starting “Means…” You have essentially created age-class specific reference ranges in this study since those were the groups with statistically significant differences – and I would clarify that in the abstract. 

A: Considering that the main goal of my study was to establish reference intervals for wild individuals of the species, I believe this information would come before the variations found among different categories. Besides, not much differences were found among the groups, with exception of WBC. For these reasons I believe the results would be better present with the information about the RI coming before any variation found. 

Lines 38-39: I recommend deleting “as a result of the still-developing immune system.” There are other potential explanations for calves having higher WBCs (higher lung worm burden in younger animals - which has been observed in other dolphin populations, or higher stress level than older animals). Given these alternate possible explanations and the fact that this is not a common finding in all populations of dolphins, I do not recommend being so definitive in explaining its cause. 

A: Attended

Introduction:

Line 52: Change “accessed” to “assessed”

A: Attended

Line 55: Recommend changing “health status approaches” to “health assessments”

A: Attended

Line 56-57: All these studies can be referred to as “capture-release” studies, I am unclear why the distinction is being made for the St. Aubin study as “netted wild population” vs capture release programs. Please clarify.

A: I meant that commonly the dolphins are captured in the coast (coastal species) many times along the year.; and in one occasion the capture happened in open ocean on open ocean species. I did some modification on the text, please check if it is appropriate now. 

Line 57: Change semicolon to comma and “parameters” to “parameter” and “baseline” to “baselines”

A: Attended

Lines 58-62: This sentence is confusing the way it is written, please reword. 

A: Attended

With reference #11 – there had not been a baseline study prior to the oil spill on that particular population (although baseline studies existed for other Tt populations that were used for comparison). I would be more explicit in explaining how this particular citation makes the authors’ point. 

A: The sentence was re written.

I also think the reference to seasonal variation in the Sarasota dolphins is confusing since you do not address seasonality at all in the remainder of the paper (even though it is controlled for with your sample timing). 

A: The sentence regarding seasonality and the reference Hall et al (2007) were removed from this part of the manuscript.

Line 64: delete “aged”

A: Attended

Line 65: This is not necessarily a common finding among the capture-release sampled dolphins in various locations. Given the age of this citation, I would recommend either striking this comment or finding additional more recent citations to make a stronger case.

A: Nabi et al (2017) also found higher WBC in younger Yangtse finless porpoises “In terms of lymphocyte and WBCs counts, age class and reproductive states were a significant variable: both were significantly higher in either calves or juveniles compared to adults in both populations.” This reference was added to the sentence. 

Line 67: Change “alteration on” to alterations in”

A: Attended

Line 69: change “englobe” to “involve”

A: Attended

Line 70: change “may reflect” to “may be reflected in”

A: Attended

Line 71: change “gender” to “sex”

A: Attended

Line 73: I believe you mean “before establishing baseline blood parameters.” Ideally you are establishing these reference ranges with animals that are minimally stressed and removing the outliers from your “normal” sampling population.

A: “baseline” was added to the sentence. Outliers were removed from the statistical analysis. 

Line 76: include citation for the most recent IUCN report

A: Attended

Line 76-77: Recommend changing “under intentional catch” (which is vague) to “actively fished commercially”

A: “actively fished” was added to the sentence. The botos are not being sold so I did not include the word “commercially”. 

Line 78: Recommend changing “leading to significant populations decrease” to “leading to a significant population decline”

A: Attended

Lines 83-84: delete comma after “reserve” and replace with “in the”

A: Attended

Line 87: Why single out the ESR here? Seems unnecessary – either please justify or delete.

A: The sentence was deleted.

Materials and Methods:

Line 109: Recommend spelling out the RDSM acronym in the figure caption

A: Attended

Line 116: Change “proximities of research floating station” to “proximity of the floating research station”

A: Attended

Lines 116-117: Recommend changing the sentence starting with “It varied” to: “The number and behavior of encircled animals varied according to environmental conditions.”

A: Attended

Line 118: If you are going to indicate the depth of the channel it would also be helpful to know the depth of your capture net (s).

A: Information included 

Lines 119-121: You state that the usual method was to seine the dolphins to the shoreline. That leads me to believe that there were other methods used to capture the dolphins in some instances. If so, please describe those methods as well, if not, please clarify the text here.

A: There are no other methods, the sentence was rewritten. 

Line 126: What were your criteria for determining pregnancy? Identification of a fetus? Recommend explicitly stating that. 

A: Yes, identification of a fetus. Information included. 

Line 139: Did you actually use a catheter or a needle? Was the blood drawn directly into the EDTA vacutainer tube or was it drawn into a syringe and then transferred to the tubes? The way this is written, it sounds like the blood from the EDTA tubes was transferred to a tube with no anticoagulant and allowed to clot, which is confusing. Please clarify your methods.

A: I have used a catheter and the blood was drawn into a syringe and then transferred to EDTA tubes. The sentence was modified. 

Line 146: How were these manual methods standardized? Were they consistently done by a single trained individual or were multiple individuals trained on standardized methods?

A: I did perform all the blood analyses. The sentence was modified. 

Line 171: For stress levels 1 & 2 behavior, perhaps you mean no/little excessive response to external stimuli? Or appropriate response to external stimuli? Relaxed dolphins should be alert and responsive to external stimuli, but not hyper-responsive. By immobilization do you mean restraint? Recommend re-wording these behavioral descriptions. 

A: Thank you for your observations. The behavioral descriptions were re-worded. 

Line 175: I think you mean “the neutrophil to lymphocyte ratio” ? 

A: Yes, thank you.

Lines 176-177: 12 individuals in each category? Are samples from the same individual compared between the two time categories or are the comparisons done between different animals? Please clarify the methods here. 

A: Comparison done between different animals, and yes 12 on each category (24 animals in total). Please let me know if it is clearer now. 

Line 177: insert ‘of’ between ‘chronology’ and ‘acute’

A: Attended

Line 181: Recommend first discussing the tests for significant differences in age class, reproductive status, and sex, then discuss the methods for creating reference ranges based on those outcomes. Were all the data tested for normality (Gaussian distribution)? Especially with some of the parameters that had smaller sample size, normal distribution cannot be assumed and should be tested.

A: I believe the establishment of the reference ranges is the most important result of this study, that´s why is comes before the differences found between the categories.

As I stated before, the normality was first observed by histogram graphs, outliers were identified and excluded from the analysis. This have caused a subsequent adjustment of the Gaussian curve. 

Line 200: Strength of correlation (weak, moderate, or strong according to r value) should be stated here. You can find references in the literature to describe correlation strength. 

A: Attended

Line 195: How were multiple comparisons controlled for in order to prevent Type I error (false positives)? You compared a lot of variables and should either control for these multiple comparisons or justify why you did not do so. 

A: What I meant here is that not all comparisons were made, for example, male (n = 2) vs female calves (n = 4) for hemoglobin, MCHC, …Type I error was prevented by the observation of the histograms and Gaussian curves. If you believe these tests were not sufficient to make an accurate analysis and If you have any suggestion on how I should proceed I would appreciate to hear. 

Line 214: Since Table 4 is cut off, I cannot see if you stated 90% RI for the calves. I am assuming you did not due to small sample size. I would explicitly state what you are presenting for the calf data here and why there are not reference ranges established. I would also recommend removing the moderate and severely stressed dolphins from the reference range calculation. 

A: I am sorry for this table layout, but the Journal requires it goes like this for review. If the Journal allows me, I can send you the table 4 in a different format. 

The calves were not included on the reference interval because they presented statistically significant higher WBC than adults and subadults. As you said, I did not calculate RI for calves due to the small sample size, but I did include the min and max values. 

I have included in the text the explanation of why blood values from calves are in a different column. 

Line 220: Formatting in the version I have cuts off the left (parameter) and right (last column is calves min) margins of this table. Cannot fully evaluate data presented. Recommend including more details regarding the reference values in the table caption (90% double sided reference intervals).

A: Same situation as previous comment. 

Line 221: In the methods, it stated the level of significance was set at p < 0.05 but here it states p < 0.001. Why the discrepancy?

A: Because the minimum of significance was 0.05, but here it was found as 0.001. 

Line 234: In Fig 2: Recommend adding sample sizes (n) above each age class box plot 

A: Number of each age class was included in the figure caption.

Line 242: Again, recommend including sample sizes in Fig 3 (either in the caption or in the legend)

A: Number of each age class was included in the figure caption.

Line 251: I do not see medians in table S1, delete reference to them here.

A: Thank you for your observation, medians was deleted.

Line 252: These are not reference intervals, recommend deleting ”RI” and just referring to the data in table S1 as summary statistics. 

A: Thank you for your observation, RI was deleted.

Line 263: Fig 4: Recommend including r and p values either on the figure or in the caption. Same comment for Figs 5 and 6 as well. 

A: Attended

Line 254: Recommend spelling out cardiac rate and respiratory frequency in this section title as these are not common abbreviations. 

A: Attended

Line 268: Correlation should be categorized as weak, moderate or strong based on the r values. 

A: Attended

Discussion:

Line 300: change “comes” to “come” – “data” is a pleural noun (singular “datum”)

A: Attended

Line 301: change to “in the 1970’s”

A: Attended

Lines 304-310: Captive boto were also likely less stressed than the wild-caught boto, which would also affect the WBC count. Subclinical bacterial, fungal, protozoal, and viral infections should also be included in the differentials for elevated WBC in wild vs managed populations.

A: I agree with you. These information were added.

Line 320: change “an” to “a” Also, without demonstrating a correlation between ESR and inflammation, I would caution against evaluating this parameters utility here. While the narrow range MAY indicate that this could be a useful indicator IF it does in fact change with inflammation, you have not proven the latter yet.

A: Attended. 

I agree and the text was modified. 

Lines 318-320: Was there any correlation between ESR and WBC count? If not, this makes it more likely that the WBC/neut elevations were due to stress rather than inflammation/infection/antigenic stimulation.

Lines 321-330: Also need to include stress as a differential for higher WBC counts in juveniles. And a note that this is not a consistent finding among all cetacean populations. 

Line 335: change “was” to “were”

A: Attended

Line 340: Change “towards” to “on”

A: Attended

Line 342: Can you present data in the results regarding the sample sizes according to each trimester if available to help elucidate this point?

Line 343: change “values” to “value” and “towards non-pregnant females” to “between pregnant and non-pregnant females”

A: Attended

Lines 356-358: Due to the spleen’s small size relative to the dolphin’s overall body mass, my understanding is that unlike in other species, splenic contraction is not considered a significant differential for elevated RBC, HCT, Hb in cetaceans. 

A: Thank you for your observation. Another explanation was presented as a potential source of Hb, HCT and RBC rise in dolphins under acute stress. 

Line 359: Recommend using the term “epinephrine” rather than “adrenalin”

A: Attended

Line 369: change “access” to “assess”

A: Attended

Line 373-374: Also cannot rule out subclinical pulmonary disease as contributing to this trend, and this should be included as a differential for that finding. 

Lines 376-378: Please reword this sentence for clarity.

Line 379: Add “can” between ‘animals’ and ‘have’ – as this is not always a consistent finding in cetaceans

A: Attended

Lines 384-394: More clarification in your discussion of the different types of stress leukograms is warranted – since you have the expected neutrophilia with a corticosteroid stress response, but not the associated lymphopenia (as with the “typical stress leukogram” that you cite). Lymphocytosis would be associated with an epinephrine-induced response. While you mention both of these hormone-induced pathways, your explanation of the lymphocytosis is incomplete.

Line 400: change ‘access’ to ‘assess’

A: Attended

Line 403: change “ad” to “and”

A: Attended

---

## [Editor Report · Decision Letter 1]

11 Dec 2019

Hematologic profile of Amazon river dolphins Inia geoffrensis and its variation during acute capture stress

PONE-D-19-17847R1

Dear Dr. Drummond de Mello,

We are pleased to inform you that your manuscript has been judged scientifically suitable for publication and will be formally accepted for publication once it complies with all outstanding technical requirements.

With kind regards,

Susana Caballero, PhD

Academic Editor

PLOS ONE

Additional Editor Comments (optional):

Thank you for working on all the comments and suggestions raised by the reviewers
---

## [Editor Report · Acceptance letter]

16 Dec 2019

PONE-D-19-17847R1 

Hematologic profile of Amazon river dolphins Inia geoffrensis and its variation during acute capture stress 

Dear Dr. Mello:

I am pleased to inform you that your manuscript has been deemed suitable for publication in PLOS ONE. Congratulations! Your manuscript is now with our production department. 

With kind regards,

on behalf of

Dr. Susana Caballero 

Academic Editor

PLOS ONE